## Enhancement of near-inertial waves by cyclonic eddy in

# the northwestern South China Sea during spring 2022

- Qi'an Chen<sup>1,2</sup>, Hongzhou Xu<sup>1\*</sup>, Dongxiao Wang<sup>3,4</sup>, Bo Hong<sup>5\*</sup>, Chunlei Liu<sup>6,7,8</sup>,
- Zheyang Zhang<sup>1</sup>, Huichang Jiang<sup>1</sup>, Wei Song<sup>3</sup>, Tong Long<sup>1</sup>, Ling Wang<sup>1,2</sup>, Sumin Liu<sup>1</sup>,
- Rongjie Chen<sup>1,2</sup>
- <sup>1</sup>Institute of Deep-sea Science and Engineering, Chinese Academy of Sciences, Sanya, 572000, China
- <sup>2</sup>University of Chinese Academy of Sciences, Beijing, 100049, China
- <sup>3</sup>School of Marine Sciences, Sun Yat-sen University, Zhuhai, 519000, China
- <sup>4</sup>South Marine Science and Engineering Guangdong Laboratory (Zhuhai), Zhuhai, 519082, China
- <sup>5</sup>School of Civil and Transportation Engineering, South China University of Technology, Guangzhou,
- 510641, China
- <sup>6</sup>College of Ocean and Meteorology, Guangdong Ocean University, Zhanjiang, 524088, China
- <sup>7</sup>South China Sea Institute of Marine Meteorology, Guangdong Ocean University, 529568, Zhanjiang,
- China
- 8CMA-GDOU Joint Laboratory for Marine Meteorology, Guangdong Ocean University, Zhanjiang,
- 524088, China
- Correspondence to: Hongzhou Xu (Email: hzxu@idsse.ac.cn); Bo Hong (bohong@scut.edu.cn)
- Abstract. By analyzing datasets derived from four moorings during spring 2022, this study provides
- direct evidence that near-inertial waves (NIWs) can be largely enhanced by a passing cyclonic eddy
- (CE) in the northwestern South China Sea. Results show that the enhancement of NIWs mainly
- occurred at north side of the CE due to asymmetry of eddy structure. In vertical, the enhancement
- concentrated at above 200 m and reached peaks at around 100 m. Significant energy transfer rates
- between the CE and NIWs appeared at the same depth of the enhancement can reach a magnitude of
- 10<sup>-8</sup> m<sup>2</sup>/s<sup>3</sup> at the CE's edge. Under the impact of the CE, power of the first five NIWs modes was
- promoted significantly and dominated by the second and third modes. Overall, the CE transferred
- energy to NIWs before near-inertial kinetic energy reaching its peaks, while NIWs gave energy back to
- the CE after the peaks.

## 1 Introduction

- Near-inertial waves (NIWs) are ubiquitous features throughout the global ocean with frequencies
- near the Coriolis frequency f (Garrett, 2001). As dominant modes of high-frequency variability in
- oceans, they contain half of the kinetic energy in internal wave fields (Alford, 2003; Alford et al., 2016;

Ferrari and Wunsch, 2009). NIWs transfer energy from mixed layer to interior and ultimately dissipate 33 into microscale turbulence, providing an energy source for abyssal diapycnal mixing (Chen et al., 2017; Ferrari and Wunsch, 2009). Therefore, NIWs are of vital importance for energy cascade among 34 35 multiscale dynamic processes. 36 Due to horizontal spatial scales of 10-100 km and slow group speed of NIWs, they are likely to interact strongly with mesoscale eddies in oceans (Alford et al., 2016). Besides resonant frequency 37 38 shifting from local f to the effective inertial frequency caused by mesoscale vorticity (Klein et al., 2003; 39 Kunze, 1985; Weller, 1982), energy exchange between eddies and NIWs also plays an important role in oceanic energy cascade (Ferrari and Wunsch, 2009; Thomas, 2017). Researchers have stated that the 40 41 NIWs can extract energy from eddy and affect the vertical material transport (Barkan et al., 2021; 42 Esposito et al., 2023). Moreover, energy transfer from mean flows can balance the dissipation of 43 near-inertial energy near the critical layer, thereby conserving near-inertial energy during eddy 44 migration. (Xu et al., 2022b). 45 Jing et al. (2017) and Jing et al. (2018) suggested that there is a permanent energy transfer from 46 eddies to NIWs under a positive Okubo-Weiss parameter condition. Furthermore, Yu et al. (2022) 47 revealed that enhanced near-inertial kinetic energy (NIKE) is found preferentially in regions of 48 anticyclonic vorticity. Using surface drifter dataset, Liu et al. (2023) indicated that bidirectional energy 49 transfer exists between eddy and NIWs in the global oceans. Above studies all emphasized role of eddy 50 on affecting frequencies of NIWs, NIKE as well. 51 As the largest semi-enclosed marginal sea in the northwestern Pacific Ocean, the South China Sea 52 (SCS) has frequent eddy activities (Chen et al., 2011; Chu et al., 2014; Nan et al., 2011; Wang et al., 53 2008; Wang et al., 2003). Early studies, by using various eddy detection algorithms, have statistically 54 characterized the mean properties of eddies, identifying peak occurrences along western boundary 55 currents and shelf break regions (Lin et al., 2007; Wang et al., 2003; Xiu et al., 2010). Specifically, 56 Chen et al. (2011) indicated that eddies occur 35%-60% of the time in the northern SCS, underscoring 57 their critical role in regional oceanography. Recent advancements in remote sensing and in-situ 58 observational technologies have enabled significant insights into eddy formation and their 59 three-dimensional structures (Chu et al., 2022; He et al., 2018; Nan et al., 2011; Wang et al., 2023; 60 Zhang et al., 2016). Due to their unique three-dimensional structures and spatiotemporal scales, eddies mediate significant material transport (He et al., 2018; Zhang et al., 2015; Zhang et al., 2019) and 61

substantial energy exchanges with internal waves and western boundary currents (Chu et al., 2014; Fan et al., 2024; Huang et al., 2018; Liu et al., 2023; Xu et al., 2022a; Zhang et al., 2023; Zhao et al., 2023).

In the northwestern SCS, large portion of eddies propagate westward and terminate near Xisha Trough (Fig. 1a)(Wang et al., 2003; Zhai et al., 2010), making this place as an ideal area for investigating NIWs and its interaction with eddies. However, interaction and energy exchange processes between them remain to be investigated in this area. In this study, four moorings, each equipped with a Teledyne RD Acoustic Doppler Current Profiler (ADCP), were deployed in the Xisha Trough of the northwestern SCS (Fig. 1a). During spring 2022, a cyclonic eddy (CE) propagated westward across the mooring array, giving an opportunity to study energy transfer between NIWs and the CE in this area. We introduce data and methods in Section 2, present the observation results in Section 3, discuss energy transfer between NIWs and the CE in Section 4, and give a conclusion in Section 5.

**Figure 1: (a)** Location of four moorings (Q1-Q4) in the northwestern SCS. The shaded background color represents topography. Orange lines indicate tracks of long-lasting eddy propagating from the Luzon area to the Xisha Trough from 1993 to 2023. **(b)** Structure of upward-looking ADCP mooring along the slope of topography. **(c)** The power spectra during eddy period (blue line) and whole observation period (black line). The spectra are

79 averaged above 200 m.

## 2 Methodology

## 81 **2.1 Data**

The four moorings (Q1-Q4) were deployed in the study area on August 21-23, 2021. One upward-looking 75-kHz ADCP along with a CTD (SBE 37sm) was fixed at approximately 480 m depth and continuously monitored current velocity for each mooring (Fig. 1b). The ADCPs were set to have 30 bins with 16 m of vertical interval and 30 minutes of temporal interval, enabling extraction of NIWs of the upper ocean. The moorings were recovered on November 13-15, 2022. Several ADCP bins near the sea surface were omitted due to large fluctuations, and the remaining were vertically linearly interpolated to 1m resolution. Due to the significant vertical fluctuations based on CTD data in certain periods, flow velocity compensation correction was applied to ADCP data, which was calculated based on depth change and flow direction. The Copernicus Marine Environment Monitoring Service (CMEMS) provides daily geostrophic current and sea level anomaly (SLA) data with a resolution of 0.25° × 0.25°, which were used to detect eddy during the observation period. The 1/12° three-dimensional products were obtained from CMEMS to calculate energy transfer rate between eddy and NIWs. The eddy dataset from Archiving, Validation, and Interpretation of Satellite Oceanographic (AVISO) was used to provide the trajectories and edges of the eddy. World Ocean Atlas (WOA18) data was used to extract temperature and salinity data. The European Centre for Medium-Range Weather Forecasts (ECMWF) Reanalysis V5 (ERA-5) data, which has hourly temporal and 0.25° spatial resolutions, was used to calculate near-inertial energy input from the wind field.

## 100 **2.2 Method**

The method for ADCP velocity flow velocity compensation correction is as follows:

$$\vec{V}_{\text{true}}(t) = \vec{V}_{\text{measured}}(t) + \vec{V}_{\text{platform}}(t),$$
 (1)

$$\vec{V}_{platform}(t) = \frac{\Delta x}{\Delta t} \vec{i} + \frac{\Delta y}{\Delta t} \vec{j},$$
 (2)

$$(x,y) = (\rho\cos\theta, \rho\sin\theta),$$
 (3)

$$(\rho, \theta) = (\sqrt{L^2 - z_{top}^2}, \alpha),$$
 (4)

- where  $\vec{V}_{true}$ ,  $\vec{V}_{measured}$  and  $\vec{V}_{platform}$  are the true velocity, observed velocity and the platform
- movement velocity, respectively.  $(\rho, \theta)$  is the polar coordinate position of the ADCP with the initial
- position as the origin. L,  $z_{top}$  and  $\alpha$  are the length of the rope at the position of the ADCP, the depth
- of the ADCP and the direction angle of the horizontal projection.

- The eddy-NIW energy transfer rate  $\varepsilon$ , where a positive value indicates energy transfer from eddies
- to NIWs and a negative value denotes the reverse, can be quantitatively calculated following Jing et al.
- (2018):

$$\varepsilon = -\left(\langle u_i u_i \rangle - \langle v_i v_i \rangle\right) \frac{S_n}{2} - \langle u_i v_i \rangle S_s, \tag{5}$$

- where  $S_n = \frac{\partial u_g}{\partial x} \frac{\partial v_g}{\partial y}$  and  $S_s = \frac{\partial u_g}{\partial y} + \frac{\partial v_g}{\partial x}$  are the normal strain and shear strain of the geostrophic
- velocity  $u_g$  and  $v_g$ , which are obtained from the CMEMS three-dimensional reanalysis data (Chen et
- al., 2023). We used fourth-order Butterworth band-pass filter with the cutoff frequency (0.8f-1.2f) to
- separate near-inertial velocity  $u_i$  and  $v_i$  from the ADCP data. The  $\langle \cdot \rangle$  represents a moving average of
- 3 internal wave periods.
- Near-inertial wind work was estimated as (Alford, 2001; Dasaro, 1985; Voet et al., 2024):

$$121 W = \vec{\tau} \cdot \vec{u}_i, (6)$$

- where  $\vec{u}_i$  is near-inertial velocity at sea surface derived from CMEMS products and  $\vec{\tau}$  is wind stress
- calculated as (Alford, 2020; Liu et al., 2019):

$$\vec{\tau} = \rho_a C_D |\vec{U}_{10} - \vec{u}_c| (\vec{U}_{10} - \vec{u}_c),$$
 (7)

- where  $\rho_a$ =1.3kg/m<sup>3</sup> is density of air,  $\vec{U}_{10}$  is 10-m wind velocity vector derived from ERA5 data,  $\vec{u}_c$
- is ocean current vector derived from CMEMS data,  $C_D$  is drag coefficient (Oey et al., 2006).
- We used the Okubo-Weiss parameter (OW) (Provenzale, 1999) method to detect the CE. The OW
- parameter was computed from the horizontal velocity field as follows:

$$\sigma = S_{\rm sh}^2 + S_{\rm st}^2 - \zeta^2, \tag{8}$$

- where  $S_{\rm sh} = \frac{\partial u}{\partial y} + \frac{\partial v}{\partial x}$  and  $S_{\rm st} = \frac{\partial u}{\partial x} \frac{\partial v}{\partial y}$  are the shear and strain deformation, respectively, and  $\zeta$  is the
- relative vorticity.

## **3 Results**

## 3.1 Near-inertial frequency and NIKE

The snapshots of SLA and surface geostrophic velocity fields show that the CE approached the mooring array on February 1 (Fig. 2). Its center passed through Q3 around February 26 and it left the mooring array on March 9. During the entire observation period, NIWs exhibited a small blue shift of near-inertial frequency with a peak value of 0.616 cycles per day (cpd) (Fig. 1c), while a significant blue shift occurred during the eddy period (February 1 - March 9) due to the background positive vorticity of the CE. The peak of spectral frequency ( $\omega_p$ ) reached 0.667cpd, with a relative frequency shift ( $\frac{\omega_p - f}{f}$ , RFS) of about 10.8% in this area. It is significantly larger than the global RFS (Guo et al., 2021), suggesting significant impact of the CE on local near-inertial frequencies. Meanwhile, we calculated the relative vorticity based on surface geostrophic currents and the local inertial frequency, and found that the average spectral peak frequency reached about 0.69 cpd during the eddy period. This value is very close to our observed  $\omega_p$ .

Figure 2: (a-j) The snapshot of sea level anomaly and geostrophic current vector during February 1-March 9. The black line indicates the Okubo-Weiss parameters.

Horizontal velocity wavelet power spectra show that strong power of NIWs occurred during eddy period but with large variations at different eddy stages (Figs. 3a-d). To quantify NIKE at different stages of the CE, we defined two time periods named as Period1 (February 16-26) and Period2 (February 26-March 8), covering 10 days before and after the eddy's center passed through the mooring array. In addition, no eddy period with calm wind during June 1-20 was chosen for extra comparison. Figs. 3e-h show temporal and vertical variations of NIKE at the four moorings during eddy period. The missing data of NIKE at surface layers are due to mooring swing caused by strong currents. It can be seen that NIKE was enhanced largely at above 200 m during Period2. And peak values of NIKE concentrated at around 100 m. The time-averaged NIKE during Period2 has almost an order of magnitude larger than that during the no eddy period (Figs. 3i-l), suggesting significant impact of the CE on local NIKE. Moreover, NIKE illustrates asymmetry in space during eddy period (Figs. 3e-l). At north side of the CE (Q1-Q2), NIKEs became much stronger than that at south side of the CE (Q3 and Q4), especially at Q2 with a maximum value up to 12.0 J/m³ (Fig. 3f), which has the same magnitude as the result observed by Xu et al. (2022b).

**Figure 3: (a-d)** 100 m-depth horizontal velocity wavelet power spectra at Q1-Q4 during eddy period. **(e-h)** Vertical distribution of NIKE at Q1-Q4. The gray triangles indicate the time when the CE edge contacts and leaves

the mooring array. (i-l) Vertical distribution of time-averaged NIKE during 'no eddy period' (black line), 'Period 1' (blue line), and 'Period 2' (red line) at Q1-Q4. The red dashed lines mark two periods of the CE.

## 3.2 Impact of eddy on NIWs

To ensure the fact that the CE could largely enhance NIWs in the northwestern SCS, we compared time series of vertical-integrated (above 200m) NIKE at each mooring with mooring-eddy distance, wind-input NIKE and eddy kinetic energy (EKE) in the study area (Fig. 4 and Fig. 5). NIWs were enhanced gradually accompanied by weakening EKE during Period1 when wind-input NIKE was relatively stable and minor. The averaged wind work during the eddy period is about 0.03mW/m², which is nearly two orders of magnitude smaller than that affected by typhoons (Ouyang et al., 2022; Yuan et al., 2024) and several times smaller than the wind work results of Voet et al. (2024) in the Iceland basin. Results suggest a vivid energy transfer from Eddy to NIWs during this period that will be quantified and discussed in Section 4. After passing of the CE's center, the CE enhanced NIWs faster, indicating more energy transfer from eddy to NIWs during Period2 than that during Period1. The slight increase of EKE during Period2 was contributed by background western boundary current velocity input (Fig. 2).

**Figure 4:** Time series of raw (thin lines) and daily-smoothed (thick lines) depth-integrated (above 200 m) NIKE at Q1-Q4 (solid lines) accompanying with distance between the CE's center and each mooring (dashed lines) during eddy period. The black dashed line represents the mean distance between the CE's center and four moorings. The triangles mark the time when the eddy edge contacts and leaves the mooring array.

**Figure 5: (a)** Time series of raw (thin lines) and daily-smoothed (thick lines) averaged wind-input NIKE at Q1-Q4. **(b)** Time series of raw (thin lines) and daily-smoothed (thick lines) area integrated EKE.

 In space, NIWs intensity reached 600 J/m<sup>2</sup> at Q2, whereas values at Q3 and Q4 were approximately 300 J/m<sup>2</sup>. This difference may be associated with the CE's vorticity distribution, where relative vorticities observed north of the CE (between Q1-Q2) were notably stronger than those measured south of the CE (between Q3-Q4) (Fig. 6). Zhao et al. (2021) and Zhao et al. (2023) pointed out that NIWs generation is significantly influenced by the eddy structure in which eddy with stronger shears tend to generate more powerful NIWs. In our case, Fig. 2 shows that the north side of the CE exhibited spatial overlap with the western boundary current of the northwestern SCS (Fig. 2) where strong shear could enhance NIWs largely at this side (Zhao et al., 2021, 2023).

Figure 6: Time series of raw (thin lines) and daily-smoothed (thick lines) relative vorticity among four moorings.

Vertical gray dashed lines mark two periods of the CE.

## 3.3 Impact of eddy on near-inertial modes

The CE not only affected frequency and energy of NIWs, but also their modes in the study area. Here, we calculated vertical-averaged energy (above 200 m) of the first five modes of near-inertial velocity during eddy period (Figs. 7a-d). All five modes grew since Period1 and reached peak values at Period2. But the CE had different influence on different modes with low modes (mainly the second and third modes) being significantly enhanced, especially near the CE's center. Low modes rose from 46% (Q2) and 54.4% (Q3) of total energy during Period1 to 87.6% (Q2) and 79.5% (Q3) during Period2, respectively (Figs 7e-f). The first mode has longer vertical wavelength and propagates faster than other modes that make it easy escape from eddy's influence (Chen et al., 2013). Overall, energy proportion of the first five modes were promoted from 81.5%, 66.8%, 69%, and 78.5% during Period1 to 90.8%, 88.8%, 87.7%, 94% at Q1-Q4, respectively (Figs. 7e-f). In conclusion, the cyclonic eddy has a prominent influence on different modes of NIWs, resulting in the intensification of lower-mode energy within its core region, especially for modes 2 and 3.

**Figure 7:** (a-d) Time series of NIKE for the first five modes at Q1-Q4 during eddy period. (e-f) The time-averaged proportions of each mode and all five modes during period 1 and period 2 at Q1-Q4. Black border indicates proportion sum of the first three modes.

## 4 Discussion

Energy exchange between eddies and NIWs is one of the most important processes in oceanic energy cascade (Alford et al., 2016; Ferrari and Wunsch, 2009; Thomas, 2017). By simulations, Jing et al. (2017) found that the energy transfer efficiency from eddies to NIWs is about 2% of the near-inertial energy input by the wind in the Kuroshio Extension region. Based on surface drifter dataset, Liu et al. (2023) stated that energy transfer efficiency can reach about 13%, indicating previous underestimation of eddy impact on NIWs. For obtaining precise results, direct ocean current measurement by long-term mooring is essential (Jing et al., 2018). In this section, eddy-NIWs energy transfer rate during eddy period in the study area is qualified and discussed (Fig. 8). Corresponding to the layer of NIKE enhancement (Fig. 3), large energy transfer rates occurred at above 200 m with the peak values at around 100 m during eddy period, rather than surface and mixing layers (Jing et al., 2017; Liu et al.,

2023). Both positive and negative transfer rates can reach a magnitude of 10<sup>-8</sup> m<sup>2</sup>/s<sup>3</sup> in the study area, in which they are of same order of magnitude as the results reported by Chen et al. (2023) in the Northwestern Pacific Ocean, but they are significantly stronger than the results of Jing et al. (2018) in the Gulf of Mexico. The differences may be attributed to the strength of eddies, their rotation direction and the intensity of NIWs. Previous studies have shown that eddy rotation plays a critical role in energy transfer and NIWs propagation due to differences in vorticity input and stratification modulation (Alford et al., 2016; Jing et al., 2017). In addition, eddy-NIWs energy transfer is largely dependent on eddy structure in which high rate can be caused by strong eddy shear (Zhao et al., 2023). It can be found that energy transfer at Q1 and Q4 were more active than that at Q3 due to occurrence of large shear strain at CE's edge (Figs. 9a-9b). Although NIKEs at Q4 were relative weak, the strong shear strains of the low frequency flow promoted local transfer rates at this area. Thomas and Daniel (2020) and Li et al. (2022) both stated that NIWs draw energy from the background flow when the energy of NIWs is small compared to that of the background flow, and release energy to the background flow when the energy of NIWs is large compared to that of the background flow. Similarly, our results show that positive/negative energy transfers from the CE to NIWs dominated "Before Strongest"/ "After Strongest" periods at most mooring stations (Fig. 9c), indicating an inhomogeneous and bidirectional energy transfer between eddy and NIWs during different periods in the northwestern SCS. Moreover, we compared the energy transferring from the CE to the NIWs with net NIKE increasement during "Before Strongest" period. Here, we multiplied the net energy transfer rate by density, then integrated them over the 0-200 m depths during "Before strongest" period as:  $\int_{T_{strongest}-7 days}^{T_{strongest}} \int_{-200m}^{0m} \rho_0 \overline{\epsilon} \, dz dt$ , in which ε represents net energy transfer rate. Results show that there were 79 J/m<sup>2</sup>, 346 J/m<sup>2</sup>, 47 J/m<sup>2</sup>, and 115 J/m<sup>2</sup> energy transferring from the CE to the NIWs during the period at Q1-Q4, respectively, which account for about 16%, 88%, 27%, and 47% of the net NIKE increasements (492 J/m<sup>2</sup>, 394 J/m<sup>2</sup>, 175 J/m<sup>2</sup> and 245 J/m<sup>2</sup> at Q1-Q4, respectively). In average, they account for approximately 45% of the net NIKE increasement (325 J/m<sup>2</sup>) in this area indicating a key role of mesoscale eddy on NIWs in the Northwestern SCS.

Figure 8: (a-d) Vertical distribution of energy transfer rate between the CE and NIWs at Q1-Q4 during eddy period.

Figure 9: (a) Time series of depth-averaged (above 200 m) normal strain  $(-(\langle u_i u_i \rangle - \langle v_i v_i \rangle) \frac{S_n}{2})$  of the CE for each mooring. (b) Time-series of depth-averaged (above 200 m) shear strain  $(-\langle u_i v_i \rangle S_s)$  of the CE for each mooring. (c) Time- (7 days) and depth-averaged (above 200 m) positive (light red), negative (light blue) and net transfer rate (dark red and blue indicate positive and negative values) before and after the NIKE reaching its peak at Q1-Q4.

## **5 Conclusion**

During spring 2022, the CE passed through the northwestern SCS. Our four long-term moorings with ADCP instruments captured the interaction and energy exchange processes between eddy and local NIWs for the first time in this area. We found that NIWs can be largely enhanced by the passing CE. Horizontally, the CE transferred more energy to NIWs at the north side than that at the south side of the CE. This disparity may be attributed to the eddy asymmetry with stronger relative vorticity and shear in the northern region, which can significantly amplify NIWs generation (Zhao et al., 2021, 2023). In vertical, the enhancement of NIKE occurred at above 200 m, with a maximum exceeding 12 J/m³ at a depth of 100 m. Power comparison of different NIWs modes during eddy period indicate that the CE

- promoted percentage of first five modes, especially the second and third modes. Overall, NIWs drew
- energy from the CE during the enhancing period of NIKE, while they gave energy back to the CE
- during weakening period of NIKE. This study is helpful for understanding multi-scale interaction and
- energy cascade in the northwestern SCS.
- Data availability. The CMEMS products are available at
- https://data.marine.copernicus.eu/product/GLOBAL MULTIYEAR PHY 001 030/services. The
- ERA-5 wind data are available at https://doi.org/10.24381/cds.adbb2d47. The WOA2018
- climatological monthly mean mixed layer depth data are available at
- https://www.ncei.noaa.gov/access/world-ocean-atlas-2018/. The AVISO eddy data are available at
- <a href="https://www.aviso.altimetry.fr/en/data/products/value-added-products/global-mesoscale-eddy-trajectory">https://www.aviso.altimetry.fr/en/data/products/value-added-products/global-mesoscale-eddy-trajectory</a>
- -product.html. The bathymetric data is from GEBCO Gridded Bathymetry Data
- (https://download.gebco.net). The data used for plotting Figures for this paper is available at:
- <u>https://doi.org/10.5281/zenodo.15705180</u>.
- Acknowledgments. This research is supported by Hainan Provincial Natural Science Foundation of
- China (Grant No. 423RC547), the National Natural Science Foundation of China (Grant No. 42376023,
- 42176033), the Natural Science Foundation of Guangdong Province (Grant No. 2024A1515012218 and
- 2022A1515011736), the Innovational Fund for Scientific and Technological Personnel of Hainan
- Province (Grant No. KJRC2023D39), and the Youth Innovation Promotion Association CAS (Grant No.
- 2022373).
- Author Contribution. H.Z.X. and B.H. conceived the central idea. Q.A.C. and H.Z.X. conducted most
- of the analyses and generated the Figures. Q.A.C., B.H. and H.Z.X. wrote the main manuscript. D.X.W,
- C.L.L, Z.Y.Z., H.C.J. contributed to the revision of the manuscript. H.Z.X., Z.Y.Z., H.C.J., W.S., T.L.,
- 297 L.W., S.M.L. and R.J.C. conducted observations of the Mooring Array and participated in data analysis.

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

<sup>425</sup> 

<sup>426</sup>