# Peer review of "the northwestern South China Sea during spring 2022"

_EGUsphere, 2025_

## Author Comment (AC1)

MS No.: egusphere-2025-283
Title: Enhancement of near-inertial waves by cyclonic eddy in the northwestern South China Sea during spring 2022

**Point-by-Point Response to Reviewers**

In this point-by-point response, we reproduced the comments (**black font**), provided our responses (**blue font**), and highlighted the corresponding revisions in **red**.

**Responses to Reviewer #2**

1. This paper presents mooring ADCP observations in the northwestern SCS and interprets the variations in near-inertial energy based on analyses of satellite and reanalysis data. The energy exchange between mesoscale eddies and near-inertial waves is an interesting and important topic, and the authors provide observational evidence that near-inertial wave energy is enhanced in cyclonic eddies. I agree with the publication of this paper, but before it is published, I would like the authors to address some misleading statements and unclear descriptions.

**Response:** We appreciated for the reviewer's comments.

**Major comments:**

2. **L22-24:** 'Significant energy transfer rates between the CE and NIWs appeared at the same depth of the enhancement that can reach up to m2/s3 at the CE's edge.' : What percentage of near-inertial and eddy kinetic energy does this correspond to?

**Response:** Thank you for your comments. We calculated the percentage changes in the averages of NIKE and EKE for two time periods: before February 16th and from February 16th to March 8$^{th}$, which is 115.2% and -71.6% respectively. We have added the percentage changes in the averages of NIKE and EKE in section 3.2: "NIWs were enhanced gradually accompanied by weakening EKE during Period1 when wind-input NIKE were relatively stable and minor. The percentage changes in the averages of NIKE and EKE for two time periods, before February 16th and from February 16th to March 8th, were 115.2% and -71.6%, respectively. Results suggest a vivid energy transfer from Eddy to NIWs during this period that will be quantified and discussed in Section 4."

3. **L56-59:** 'Researchers ...' : This sentence is redundant and repeats information from L52-53. The authors should specifically present the findings of previous studies regarding mesoscale eddies in the SCS.

**Response:** Thanks for the comment. To make it more clear, we have revised the whole paragraph as: "As the largest semi-enclosed marginal sea in the northwestern Pacific Ocean, the South China Sea (SCS) has frequent eddy activities (Wang et al.,

2003; Wang et al., 2008; Chen et al., 2011; Nan et al., 2011; Chu et al., 2014 ). Early studies, by using various eddy detection algorithms, have statistically characterized the mean properties of eddies, identifying peak occurrences along western boundary currents and shelf break regions (Wang et al., 2003; Lin et al., 2007; Xiu et al., 2010). Specifically, Chen et al. (2011) indicated that eddies occur 35%–60% of the time in the northern SCS, underscoring their critical role in regional oceanography. Recent advancements in remote sensing and in-situ observational technologies have enabled significant insights into eddy formation and their three-dimensional structures (Nan et al., 2011; Zhang et al., 2016; He et al., 2018; Chu et al., 2022; Wang et al., 2023). Due to their unique three-dimensional structures and spatiotemporal scales, eddies mediate significant material transport (Zhang et al., 2015; He et al., 2018; Zhang et al., 2019) and substantial energy exchanges with internal waves and western boundary currents (Chu et al., 2014; Huang et al., 2018; Xu et al., 2022; Liu et al., 2023; Zhang et al., 2023; Zhao et al., 2023; Fan et al., 2024).”

4. **L86-94:** This paragraph lists the datasets used in the analysis, but it is hard to follow because the authors do not specify the variables for each dataset. Both the CMEMS and AVISO datasets include SLA data, but it is unclear which part of the analysis each of the two SLAs is used in.

**Response:** Sorry for the confusion. We have rewritten the introduction to the usage of AVISO data to “The eddy dataset from Archiving, Validation, and Interpretation of Satellite Oceanographic (AVISO) was used to provide the trajectories and edges of the eddy.”

5. **L96:** 'Eddy-NIWs energy transfer rate': Do you mean 'energy transfer rate from mesoscale eddies to NIWs'? Specify the direction.

**Response:** Here, we refer to the energy transfer rate between eddies and NIWs: a positive rate indicates energy is transferred from eddies to NIWs, while a negative rate reflects energy being transferred back from NIWs to eddies. We have rewritten the sentence to specify the direction: “The eddy-NIW energy transfer rate $\varepsilon$, where a positive value indicates energy transfer from eddies to NIWs and a negative value denotes the reverse, can be qualitatively calculated following Jing et al. (2018):”

6. **L97:** The equation (1) seems different from the equation (3b) in Jing et al. (2017).

**Response:** Sorry for this error. We have corrected the cited reference to “Jing et al (2018)”. The difference in energy transfer rate calculations between the two methods is that Jing et al. (2017) used the slab model to approximate energy transfer rates within the mixed layer, while Jing et al. (2018) employed 3D observational data to compute depth-specific energy transfer rates. The results from Jing et al. (2018) can then be vertically integrated to derive energy transfer rates across different depth intervals, yielding comparable metrics to the method in Jing et al. (2017).

7. **L119-122:** Is this relative frequency shift consistent with the cyclonic eddy's relative vorticity at the mooring site?

**Response:** Yes, the frequency shift of the spectral peak is generally consistent with the results of eddy vorticity input. The power spectra in Figure 1c are averaged across four moorings for depths above 200 m. By combining the relative vorticity from Figure 6 with the local inertial frequency, we calculated the average frequency shift during the eddy period under the same conditions, which is approximately 0.69 cpd. There are some error sources: the tilted vertical structure of eddies and the temporal variations in mooring positions relative to the eddy which introduce discrepancies in frequency shift calculations. We conclude that this result aligns with the observed near-inertial frequency shift, despite potential calculation discrepancies.

8. **L126:** 'Meridional velocity wavelet power spectra': Why is only meridional velocity used? Near-inertial waves exhibit a circular hodograph, so why not include both components of velocity?

**Response:** Considering the circularly polarized nature of near-inertial internal waves, the wavelet analysis results of zonal velocity (U) are similar to those of meridional velocity (V). They exhibit consistent variations during "Period 1" and "Period 2" (Figure R1). Since presenting an additional component wouldn't provide more conclusive insights for data analysis, and considering subplot quantity and layout, we only displayed the wavelet analysis of one velocity component. The wavelet analysis results for both velocity components are shown below.

[Figure]

Figure R1. 100 m-depth meridional (left panel) and zonal (right panel) velocity wavelet power spectra at Q1-Q4 during eddy period.

9. **L153-154:** It seems too speculative. I can't discern this from Fig.1a.
**Response:** Sorry for the confusion. The strong current in the northwestern side (western boundary current) in Fig 2 can support this point.We have rewritten the sentence to "contributed by background western boundary current (Fig. 2)".

10. **L166:** 'due to vorticity asymmetry of the CE' : This seems too definitive. It is not clear from the figures whether the differences in near-inertial energy at each mooring site are attributable to the asymmetry in vorticity.
**Response:** We agree this comment. Since the paper did not utilize model sensitivity experiments to diagnose the relation between vorticity asymmetry and NIKE, we were unable to confirm that the spatial distribution was definitively caused by vortex asymmetry. Therefore we have softened the tone and rewritten the sentence to present it as a correlative speculation here: "In space, NIWs intensity reached 600 J/m² at Q2, whereas values at Q3 and Q4 were approximately 300 J/m². This difference may be associated with the CE's vorticity distribution, where relative vorticities observed north of the CE (between Q1-Q2) were notably stronger than those measured south of the CE (between Q3-Q4) (Fig. 6)."

11. **L169-170:** 'the north side of the CE was merged with the western boundary current of the northwestern SCS (Fig. 1a)': It's not clear from Fig. 1a.

**Response:** We believe that the strong current in the northwestern side (western boundary current) in Fig 2 can explain this point. We have rewritten the sentence to "In our case, the north side of the CE exhibited spatial overlap with the western boundary current of the northwestern SCS (Fig. 2)".

12. **L170-171:** 'that generated strong shear and enhanced NIWs largely at this side': This is an inference, so the authors should avoid making definitive statements.
**Response:** To avoid making definitive statements, we have rewritten the sentence to "In our case, Fig. 2 shows that the north side of the CE exhibited spatial overlap with the western boundary current of the northwestern SCS (Fig. 2) where strong shear could enhance NIWs largely at this side (Zhao et al., 2021, 2023)."

13. **L180-181:** What is this percentage to?
**Response:** The percentage is relative to the total energy. We have revised the sentence to "Low modes rose from 46% (Q2) and 54.4% (Q3) of total energy during Period1 to 87.6% (Q2) and 79.5% (Q3) during Period2, respectively".

14. **L185-186:** 'suggesting prominent influence of eddy on near-inertial modes' : This is too vague.

**Response:** Sorry for the confusion. We have rewritten the sentence to clarify the conclusion :"In conclusion, the cyclonic eddy has a prominent influence on different modes of NIWs, resulting in the intensification of lower-mode energy within its core region, especially for modes 2 and 3."

15. **L194:** 'eddy-NIWs energy transfer efficiency is about 2%': the term 'eddy-NIWs energy transfer efficiency' is unclear, so I don't understand what the '2%' refers to.

**Response:** Sorry for the confusion. We have rewritten the sentence to "By simulations, Jing et al. (2017) found that the energy transfer efficiency from eddies to near-inertial waves is about 2% of the near-inertial energy input by the wind in the Kuroshio Extension region".

16. **L200:** 'rather than surface and mixing layers' : What is the mixed layer depth at the mooring sites?
**Response:** The GLORY dataset contains MLD data, which is about 28m at the mooring sites.

17. **L207-208:** Which terms in equation (1) explain the high rate at Q1? Please provide a similar discussion as for Q4.
**Response:** To answer this question, we recalculated two right terms, normal strain $-(\langle u_i u_i \rangle - \langle v_i v_i \rangle)\frac{S_n}{2}$ and shear strain $-\langle u_i v_i \rangle S_s$ in old Eq. (1), and plotted them in Figure 9 (Figure R2). It can be seen that shear strain caused high rate at Q1 and Q4. We have rewritten the sentence to "It can be found that energy transfer at Q1 and Q4 were more active than that Q3 due to occurrence of large shear strain at CE's edge (Figs. 9a-9b). Although NIKEs at Q4 were relative weak, the strong shear strains of the low frequency flow promoted local transfer rates at this area."

[Figure]

Fig R2. (a) Time series of depth-averaged (above 200 m) normal strain $(-(\langle u_i u_i \rangle - \langle v_i v_i \rangle) \frac{S_n}{2})$ of the CE for each mooring. (b) Time-series of depth-averaged (above 200 m) shear strain $(-\langle u_i v_i \rangle S_s)$ of the CE for each mooring. (c) Time- (7 days) and depth-averaged (above 200 m) positive and negative energy transfer rates before and after the NIKE reaching its peak at Q1-Q4. Cumulation bar represents sum of positive rates minus negative rates at Q1-Q4.

18. **L209-210:** 'with small energy ratio between them': I don't understand what the authors mean.
**Response:** Sorry for the confusion. We have rewritten the sentence to "NIWs draw energy from the background flow when the energy ratio between them is small, and vice versa".

19. **Fig. 9c:** How about showing the cumulative energy transfer from the CE to NIWs? It is hard to tell the net energy transfer from the current figure.
**Response:** Thank you for your comment. We have added the cumulation bar during the two time periods in Fig.9c (Figure R2c).

20. **L228:** 'due to asymmetry of eddy structure and strong shear' : This is too vague. The authors would be able to specify which terms in equation (1) explain the result.

**Response:** Thank you for your comments. Since we did not conduct model sensitivity experiments, we were unable to directly link the asymmetry of the eddy to the enhancement of NIKE. Our inference is based on the coincidence of stronger NIKE in the northern side and larger relative vorticity there. Additionally, the conclusion that strong shear enhances near-inertial energy is derived from the modeling results from Zhao et al. (2021, 2023). Therefore, we have revised the tone and added more information to this sentence: "Horizontally, the CE transferred more energy to NIWs at the north side than that at the south side of the CE. This disparity may be attributed to the eddy asymmetry with stronger relative vorticity and shear in the northern region, which can significantly amplify NIWs generation (Zhao et al., 2021, 2023)."

**Minor comments:**

21. **L33:** 'microscale turbulent mixing' -> 'microscale turbulence'
**Response:** It has been rewritten to "microscale turbulence".

22. **L39:** 'energy transfer between ...' -> 'energy exchange between ...'
**Response:** It has been rewritten to "energy exchange between".

23. **L41:** 'affect the vertical transport' : Do you mean 'affect the energy vertical transfer' or 'affect the material vertical transport'?
**Response:** Here, we aim to mean that NIWs affect material vertical transport by generating oscillatory divergence and vertical velocity, driving upward or downward movement of materials. We have rewritten the words to "vertical material transport".

24. **L42-43:** 'Moreover, ...' : I don't understand the sentence. Could you rephrase it for better clarity?
**Response:** We have rewritten the sentence to "Moreover, energy transfer from mean flows can balance the dissipation of near-inertial energy near the critical layer, thereby conserving near-inertial energy during eddy migration.".

25. **L44:** 'permanent energy can be transferred from eddies to NIWs ...' -> 'there is a permanent energy transfer from eddies to NIWs ...'
**Response:** We have rewritten the sentence to "there is a permanent energy transfer from eddies to NIWs…".

26. **L82-83:** 'the remaining were linearly interpolated vertically' : Does this mean that the raw ADCP vertical profiles are vertically interpolated? If so, to which vertical grid? And what is the reasoning behind this?

**Response:** Thank you. We did mean that the ADCP vertical profiles were vertically interpolated to achieve a depth resolution of 1m. This was done because we observed some significant vertical fluctuations during the observation. To better correct the data based on instrument depth variations derived from CTD pressure data, a refined depth interpolation method was applied to the ADCP dataset. We have rewritten the sentence to "Several ADCP bins near the sea surface were omitted due to large fluctuations, and the remaining were vertically linearly interpolated to 1m resolution."

27. **L83-85:** What types of corrections are applied to ADCP data?
**Response:** Thank you. We have added the specific algorithm of velocity compensation correction in the method section.

The method for ADCP velocity flow velocity compensation correction is as follows:

$$\vec{V}_{true}(t) = \vec{V}_{measured}(t) + \vec{V}_{platform}(t) \qquad (1)$$

$$\vec{V}_{platform}(t) = \frac{\Delta x}{\Delta t}\vec{i} + \frac{\Delta y}{\Delta t}\vec{j} \qquad (2)$$

$$(x, y) = (\rho\cos\theta, \rho\sin\theta) \qquad (3)$$

$$(\rho, \theta) = (\sqrt{L^2 - z_{top}^2}, \alpha) \qquad (4)$$

where $\vec{V}_{true}$, $\vec{V}_{measured}$ and $\vec{V}_{platform}$ are the true velocity, observed velocity and the platform movement velocity, respectively. $(\rho, \theta)$ is the polar coordinate position of the ADCP with the initial position as the origin. L, $z_{top}$ and $\alpha$ are the length of the rope at the position of the ADCP, the depth of the ADCP and the direction angle of the horizontal projection.

28. **L96:** 'qualitatively': Do you mean 'quantitatively'?
**Response:** It has been rewritten to "quantitatively".

29. **L99:** 'the reanalysis data' : Do you mean 'the 1/12 CMEMS three-dimensional products'? It is unclear and hard to follow.
**Response:** Sorry for the confusion. Yes, we mean the CMEMS three-dimensional products. We have rewritten the sentence to "which is obtained from the CMEMS three-dimensional reanalysis data".

30. **L101:** Is near-inertial velocity derived from mooring data or CMEMS products? Unclear.
**Response:** Sorry for the confusion. We have written the sentence to "We used fourth-order Butterworth band-pass filter with the cutoff frequency (0.8f-1.2f) to separate near-inertial velocity $u_i$ and $v_i$ from the ADCP data.".

31. **L104:** Is near-inertial velocity at sea surface derived from mooring data or CMEMS products? ADCP data do not cover the surface layer.

**Response:** Yes, the near-inertial velocity used in calculating the near-inertial wind work is derived from the CMEMS products. We have written the sentence to "where $\vec{u}_i$ is near-inertial velocity at sea surface derived from CMEMS products".

32. **L107-108:** Specify which dataset each variable is derived from.

**Response:** Thank you. We have written the sentence to "where $\rho_a$ =1.3kg/m3 is density of air, $\vec{U}_{10}$ is 10-m wind velocity vector derived from ERA5 data, $\vec{u}_c$ is ocean current vector derived from CMEMS data, $C_D$ is drag coefficient".

33. **L112:** Clarify the equations for shear and strain deformation and relative vorticity, as is done for the normal strain and shear strain in L98.

**Response:** We have clarified the equations as "where $S_{sh} = \frac{\partial u}{\partial y} + \frac{\partial v}{\partial x}$ and $S_{st} = \frac{\partial u}{\partial x} - \frac{\partial v}{\partial y}$ are the shear and strain deformation, respectively, and $\zeta$ is the relative vorticity."

34. **L132:** 'The absences of NIKE' -> Do you mean 'Missing data'?

**Response:** We have written the sentence to "The missing data of NIKE at surface layers are due to mooring swing caused by strong currents."

35. **Figure 5a:** Y-axis label: 'Energy' -> This is not energy. near-inertial wind work.

**Response:** We have modified the Y-axis label of Figure 5a to "Near-inertial wind work (mW/m2)".

[Figure]

Fig R3. **(a)** Time series of raw (thin lines) and daily-smoothed (thick lines) averaged wind-input NIKE at Q1-Q4. **(b)** Time series of raw (thin lines) and daily-smoothed (thick lines) area integrated EKE.

36. **L202:** 'in which ...' : Did Jing et al. (2018) and Chen et al. (2023) target the same area in the northwestern SCS? If not, the location information should be described.

**Response:** They did not target the same area in the northwestern SCS. We have written the sentence to "Both positive and negative transfer rates can reach a magnitude of $6 \times 10^{-10}$ m$^2$/s$^3$ in the study area, in which they are several times larger than the result of Jing et al. (2018) in the Gulf of Mexico, but they are smaller than the result of Chen et al. (2023) in the Northwestern Pacific Ocean."

**Typos and grammatical correction:**

37. **L64:** 'a' Teledyne RD

**Response:** We have rewritten the sentence to "In this study, four moorings, each equipped with a Teledyne RD Acoustic Doppler Current Profiler (ADCP), were deployed in the Xisha Trough of the northwestern SCS".

38. **L67:** 'observing results' -> 'the observation results' or 'the observed results'.

**Response:** We have rewritten the sentence to "present the observation results in Section 3"

39. **L72:** 'up-looking' -> 'upward-looking' (Same for other parts)

**Response:** We have replaced all "up-looking" with "upward-looking" throughout the entire paper.

40. **L74:** 'averaged at above 200 m' -> 'averaged above 200 m'

**Response:** We have replaced "averaged at above 200 m" with "averaged above 200 m".

41. **L88:** 'during the observing period' -> 'during the observation period'

**Response:** We have replaced "during the observing period" with "during the observation period".

42. **L110:** 'method' -> 'parameter'

**Response:** We have replaced "method" with "parameter".

43. **L118-119:** 'While a significant ...': The sentence is incomplete.

**Response:** We have rewritten and combined the sentence as follows: "During the entire observation period, NIWs exhibit a small blue shift of near-inertial frequency with a peak value of 0.616 cycles per day (cpd) (Fig. 1c), while a significant blue shift occurred during the eddy period (February 1 - March 9) due to the background positive vorticity of the CE."

44. **L127:** 'To qualify' -> 'To quantify'?

**Response:** We have replaced the word "To qualify" with "To quantify"

45. **L136:** 'asymmetry in spatial' -> 'asymmetry in space'

**Response:** We have replaced the word "asymmetry in spatial" with "asymmetry in space"

46. **L151:** 'will be qualified' -> 'will be quantified'?

**Response:** We have replaced the word "qualified" with "quantified".

47. **L165:** 'In spatial' -> 'In space'

**Response:** We have replaced the word "In spatial" with "In space".

48. **L198:** 'was' -> 'is' or 'will be' would be appropriate.

**Response:** We have replaced the word "was" with "is".

49. **L207:** 'were relative weak' -> 'were relatively weak'

**Response:** We have replaced the word "were relative weak" with "were relatively weak".

50. **L209:** 'NIWS' -> 'NIWs'

**Response:** We have replaced the word "NIWS" with "NIWs".

---

## Author Comment (AC2)

MS No.: egusphere-2025-283
Title: Enhancement of near-inertial waves by cyclonic eddy in the northwestern South China Sea during spring 2022

**Point-by-Point Response to Reviewers**

In this point-by-point response, we reproduced the comments (**black font**), provided our responses (**blue font**), and highlighted the corresponding revisions in **red**.

**Responses to Reviewer #1**

1. Utilizing a mooring array, this study investigates the interaction between a cyclonic eddy (CE) and near-inertial waves (NIWs) in the northwestern South China Sea (SCS). Results demonstrate that NIWs were significantly amplified by the passage of a cyclonic eddy. Overall, the manuscript presents a well-structured analysis of an interesting topic. While the work appears suitable for publication in Ocean Science, a few minor clarifications are requested to strengthen the paper prior to final acceptance.

Response: We appreciated for the reviewer's comments.

Specific Comments

2. Line 24: "power of the first five NIWs modes were promoted significantly and dominated by the second and third modes." Use "was promoted" for subject-verb agreement.
Response: 'were promoted' has been replaced with 'was promoted'.

3. **Line 53:** "Chen et al. (2011) suggest that eddies present 35%--60% of the time in the northern SCS..." Replace "present" with "occur" for clarity.
Response: 'present' has been replaced with 'occur'.

4. **Line 60:** "making this place as a seemly area for investigating NIWs..." "seemly" is incorrect; replace with "suitable" or "ideal."
Response: 'seemly' has been replaced with 'ideal'.

5. **Line 81–82:** "flow velocity compensation correction was applied to ADCP data, which was calculated based on depth change and flow direction. Clarify the exact method used for compensation (e.g., reference to a specific algorithm or equation).
Response: We have added the specific algorithm of velocity compensation correction in the data section as below:

The method for ADCP velocity flow velocity compensation correction is as follows:

$$\vec{V}_{\text{true}}(t) = \vec{V}_{\text{measured}}(t) + \vec{V}_{\text{platform}}(t)$$

$$\vec{V}_{\text{platform}}(t) = \frac{\Delta x}{\Delta t}\vec{i} + \frac{\Delta y}{\Delta t}\vec{j}$$

$$(x, y) = (\rho cos\theta, \rho sin\theta)$$

$$(\rho, \theta) = (\sqrt{L^2 - z_{\text{top}}^{2}}, \alpha)$$

where $\vec{V}_{\text{true}}$, $\vec{V}_{\text{measured}}$ and $\vec{V}_{\text{platform}}$ are the true velocity, observed velocity and the platform movement velocity, respectively. $(\rho, \theta)$ is the polar coordinate position of the ADCP with the initial position as the origin. $L$, $z_{\text{top}}$ and $\alpha$ are the length of the rope at the position of the ADCP, the depth of the ADCP and the direction angle of the horizontal projection.

6. **Line 82:** Does the lack of data in the upper tens of meters of the mooring data lead to errors in energy calculations and filtering?
Response: As shown in Figures 2e-h, the peak of NIKE occurs around 100 m. While, NIKEs have already significantly weakened above 100 m. Therefore, we conclude that the missing data does not lead to significant errors in the results.

7. **Line 95:** Since calculating the energy conversion rate requires gradients in the x and y directions, the mooring array is a section. How is this considered when calculating the energy conversion rate?
Response: The reviewer is right that calculating geostrophic currents in $S_s$ and $S_n$ requires gradients in the x and y directions, which cannot be satisfied by mooring observations. Therefore, we use 3D velocity data from CMEMS reanalysis (6-hourly resolution) for these calculations. Since NIKE in the equations does not involve gradient computations, we directly apply filtering to mooring-derived observations to better reflect real conditions like the method applied by Chen et al. (2023).

8. **Line 97:** "moving average of 3 internal tide periods." The analysis in the article focuses on near-inertial internal waves, not internal tides. Therefore, it should be 3 "internal wave periods".
Response: We have rewritten the sentence as 'The $\langle \cdot \rangle$ represents a moving average of 3 internal wave periods.'. 'internal tide' has been replaced with 'internal wave' as well.

9. Is this the only CE throughout the year? Or why was this particular CE selected for analysis?
Response: Thanks for raising this insightful question. This is not the only CE observed during the observation period. According to the AVISO eddy dataset, six CEs were detected near the mooring location throughout the year. Among them, four were locally generated due to flow field adjustments, characterized by short durations

or weak energy. The remaining two CEs propagated westward to the observation area, both exhibiting stronger energy compared to the others. The first of these propagating eddies lasted longer, with a trajectory perpendicular to the observation transect, and triggered the most significant vertical displacement of the mooring during the observation period. Therefore, we focused our analysis on this particular eddy.

10. **Line 116:** "near-inertial frequency with peak value of 0.616 cpd" Define "cpd" (cycles per day) upon first use.
Response: We have rewritten the sentence as 'near-inertial frequency with peak value of 0.616 cycles per day (cpd)'. Definition of cpd has been added.

11. **Line 160:** "Low modes rose from 46% (Q2) and 54.4% (Q3) during Period1 to 87.6% (Q2) and 79.5% (Q3) during Period2..." Clarify what "46%" refers to (e.g., percentage of total energy?).
Response: We have rewritten the sentence as 'Low modes rose from 46% (Q2) and 54.4% (Q3) of total energy during Period1 to 87.6% (Q2) and 79.5% (Q3) during Period2, respectively'.

12. **Line 200–201:** "The differences may be attributed to the strength of eddies, their rotation direction and the intensity of NIWs." Expand on how rotation direction (cyclonic vs. anticyclonic) impacts energy transfer.
Response: We have expanded the sentence as 'The differences may be attributed to the strength of eddies, their rotation direction and the intensity of NIWs. Previous studies have shown that eddy rotation plays a critical role in energy transfer and NIWs propagation due to differences in vorticity input and stratification modulation (Alford et al., 2016; Jing et al., 2017).'

13. **Line 206:** "NIWs draw energy from background flow with small energy ratio between them, vice versa." Rephrase for clarity (e.g., "NIWs draw energy from the background flow when the energy ratio is small, and vice versa").
Response: We have rewritten the sentence as 'NIWs draw energy from the background flow when the energy ratio between them is small, and vice versa'.

14. **Line 229:** "This study is helpful for us to understand multi-scale interaction..." Replace "for us" with "for understanding" to maintain objectivity.
Response: We have rewritten the sentence to 'This study is helpful for understanding multi-scale interaction and energy cascade in the northwestern SCS.'. 'internal tide' has been replaced with 'internal wave'.

15. **Figure 1:** Panel (c): Label the x-axis of the power spectra.
Response: Thank you. We have added the label to this figure.

[Figure]

16. "The mooring dataset used for plotting Figure for this paper are available at", the grammar is incorrect.

Response: We have rewritten the sentence as "The data used for plotting Figures for this paper is available at ……".

17. Ensure consistent use of hyphens (e.g., "near-inertial" vs. "near inertial").

Response: Thank you. The revised manuscript now consistently uses "near-inertial" throughout the text. Thanks.

18. Use "cyclonic eddy (CE)" consistently after first introduction.

Response: Following the revision, the term "cyclonic eddy (CE)" is now consistently presented with its full form and abbreviation in the abstract. For maintaining conceptual continuity and precision, the full term "cyclonic eddy (CE)" is reintroduced in the introduction, with subsequent references using "CE" only.

---

## Author Response (AR2)

MS No.: egusphere-2025-283

Title: Enhancement of near-inertial waves by cyclonic eddy in the northwestern South China Sea during spring 2022

**Point-by-Point Response to Reviewers**

In this point-by-point response, we reproduced the comments (**black font**), provided our responses (**blue font**), and highlighted the corresponding revisions in **red**.

**Specific Comments**

1. The paragraph newly added in response to my previous comment is unclear and does not appear to fully address the concern I raised. I would recommend removing it.

What I intended is that the expression "significant energy transfer" is vague, and I am concerned that the estimated value of 6 x 10^-10 m^2/s^3 may be too small to represent a meaningful contribution to near-inertial energy. This paper assumes that changes in near-inertial energy are due to energy transfer between cyclonic eddies and near-inertial waves. To support this, it is necessary to compare the estimated energy transfer rate with the changes in near-inertial energy.

In Section 4 (Discussion), the authors cite Jing et al. (2017) and Liu et al. (2023), who report that the energy transfer efficiency from mesoscale eddies to near-inertial waves can be 2 % or 13 % of the wind input. Alternatively, the authors might consider estimating what fraction of the observed near-inertial energy during Period 1 (or 2) could be attributed to the energy transferred from the eddy (or to the eddy). Would it be possible to evaluate the magnitude of the energy transfer in such a quantitative manner?

**Response:** Thanks for this comment. Since the units of energy transfer rate and nearinertial energy differ, we referred to the comparison between energy conversion rate and wind work by Jing et al. (2018) and the comparison between the time integral of wind work and near-inertial energy by Voet et al. (2024). We multiplied the net energy conversion rate by density, integrated it over the upper 200 meters and the "Before strongest" time period, and then compared it with the change in near-inertial during this period. The integration equation energy  $\int_{T_{strongest}-7days}^{T_{strongest}} \int_{-200\text{m}}^{0\text{m}} \rho_0 \overline{\varepsilon} \, dz dt$ . In addition, we found that the data of energy transfer rate was excessively time-smoothed which leaded too small values of 10-10 m2/s3, so we recalculated the results which can reach about 10-8 m2/s3. We revised the old Figures 8 and 9 accordingly (see Figures R1 and R2). After the correction, results show that there were 79 J/m2, 346 J/m2, 47 J/m2, and 115 J/m2 energy transferring from the eddy to the NIWs at the Q1-Q4 during the "Before strongest" period, respectively, which accounted for about 16%, 88%, 27%, and 47% of the net NIKE increases ( $492 \text{ J/m}^2$ ,  $394 \text{ J/m}^2$ ,  $175 \text{ J/m}^2$ , and  $245 \text{ J/m}^2$ ).

We have revised the figures and descriptions related to the energy transfer rate in the manuscript.

Figure R1: (a-d) Vertical distribution of energy transfer rate between the CE and NIWs at Q1-Q4 during eddy period.

Figure R2: (a) Time series of depth-averaged (above 200 m) normal strain  $(-(\langle u_i u_i \rangle - \langle v_i v_i \rangle) \frac{S_n}{2})$  of the CE for each mooring. (b) Time-series of depth-averaged (above 200 m) shear strain  $(-\langle u_i v_i \rangle S_s)$  of the CE for each mooring. (c) Time- (7 days) and depth-averaged (above 200 m) positive (light red), negative (light blue) and net transfer rate (dark red and blue indicate positive and negative values) before and after the NIKE reaching its peak at Q1-Q4.

**Revisions to the manuscript text:**

- (1) In the Abstract, the description of the energy transfer rate magnitude was revised to "reach a magnitude of 10-8 m2/s3".
- (2) The description in last version about "The percentage changes in the averages of NIKE and EKE for two time periods, before February 16th and from February 16th to March 8th, were 115.2% and -71.6%, respectively" was removed.
- (3) In the Discussion section, the description of the energy transfer rate was updated to reflect the corrected results: "Both positive and negative transfer rates can reach a magnitude of 10-8 m2/s3 in the study area, in which they are of same order of magnitude as the results reported by Chen et al. (2023) in the Northwestern Pacific Ocean, but they are significantly stronger than the results of Jing et al. (2018) in the Gulf of Mexico."
- (4) In the Discussion section, we revised the discussion regarding the energy comparison: "Moreover, we compared the energy transferring from the CE to the

NIWs with net NIKE increasement during "Before Strongest" period. Here, we multiplied the net energy transfer rate by density, then integrated them over the 0-200 m depths during "Before strongest" period as:  $\int_{T_{strongest}-7 days}^{T_{strongest}} \int_{-200m}^{0m} \rho_0 \overline{\epsilon} \, dz dt \,, \text{ in which } \overline{\epsilon} \, \text{ represents net energy transfer rate.}$  Results show that there were 79 J/m², 346 J/m², 47 J/m², and 115 J/m² energy transferring from the CE to the NIWs during the period at Q1-Q4, respectively, which account for about 16%, 88%, 27%, and 47% of the net NIKE increasements (492 J/m², 394 J/m², 175 J/m² and 245 J/m² at Q1-Q4, respectively). In average, they account for approximately 45% of the net NIKE increasement

(325 J/m2) in this area indicating a key role of mesoscale eddy on NIWs in the

2. I think the description of the frequency peak shift being consistent with the cyclonic eddy's relative vorticity should be included in the manuscript.

Northwestern SCS."

Response: We have added the description in the manuscript as follow: "Meanwhile, we calculated the relative vorticity based on surface geostrophic currents and the local inertial frequency, and found that the average spectral peak frequency reached about 0.69 cpd during the eddy period. This value is very close to our observed  $\omega_p$ ."

3. Unless there is a specific reason to use only the meridional component, I think it would be more natural to show the wavelet spectrum of horizontal velocity by combining the power spectral densities of the zonal and meridional components.

Response: We have renewed the wavelet analysis in Fig.3 (Figure R3) with combining the power spectral densities of the zonal and meridional components.

**Figure R3:** (a-d) 100 m-depth horizontal velocity wavelet power spectra at Q1-Q4 during eddy period. (e-h) Vertical distribution of NIKE at Q1-Q4. The gray triangles indicate the time when the CE edge contacts and leaves the mooring array. (i-l) Vertical distribution of time-averaged NIKE during 'no eddy period' (black line), 'Period 1' (blue line), and 'Period 2' (red line) at Q1-Q4. The red dashed lines mark two periods of the CE.

4. This sentence is still unclear. Do you mean that NIWs draw energy from the background flow when NIKE is small compared to the background flow?

**Response:** Yes. To make it clear, we have revised the sentence as: "NIWs draw energy from the background flow when the energy of NIWs is small compared to that of the background flow, and release energy to the background flow when the energy of NIWs is large compared to that of the background flow."

5. My suggestion was to plot the net flux at each location (Q1, Q2, Q3 and Q4), rather than separately averaging the positive and negative fluxes. The cumulation bars may not be necessary because adding the results from the four locations does not appear to be meaningful in this context.

Response: We have added the net flux at each location instead of showing the cumulation bar in Fig 9 (Figure R2).

6. Please check for typos.

**Response:** We have checked for typos, and corrected grammatical and spelling errors carefully.

7. Equation (6): I missed this point in the first review. Near-inertial wind work refers to the dot product of the near-inertial component of the wind stress and the near-inertial component of the surface current velocity (e.g., Voet et al. 2024). Please verify this in your analysis.

**Response:** Thanks for this comment, we have revised this sentence as: "The averaged wind work during the eddy period is about 0.03 mW/m², which is nearly two orders of magnitude smaller than that affected by typhoons (Ouyang et al., 2022; Yuan et al., 2024) and several times smaller than the wind work results of Voet et al. (2024) in the Iceland basin."

---

## Author Response (AR3)

MS No.: egusphere-2025-283

Title: Enhancement of near-inertial waves by cyclonic eddy in the northwestern South China Sea during spring 2022

**Point-by-Point Response to Reviewers**

In this point-by-point response, we reproduced the comments (**black font**), provided our responses (**blue font**), and highlighted the corresponding revisions in **red**.

**Specific Comments**

1. Did the authors apply a moving average to the energy transfer rate estimates? The moving-average window should be substantially shorter than 7 days, as longer windows may affect the integrated energy transfer rate value. In either case, if any kind of temporal averaging was applied, the authors should explicitly describe the method and clearly state the window length in the manuscript.

**Response:** Thanks for this comment. We didn't apply any kind of other temporal averaging during the calculation. Our previous calculation lay in this formula:

$$\varepsilon = -\left(\langle u_i u_i \rangle - \langle v_i v_i \rangle\right) \frac{S_n}{2} - \langle u_i v_i \rangle S_s$$

The  $\langle \cdot \rangle$  represents a moving average of 3 internal wave periods. The moving average needs to be applied after multiplying the variables; however we applied the moving average to individual variables before the multiplication, which thus resulted in excessive smoothing of the final results.

2. Thank you for considering the revision. However, the revised sentences do not convey the intended meaning. I recommend rephrasing as follows:

**Original:**

Meanwhile, we calculated the relative vorticity based on surface geostrophic currents and the local inertial frequency, and found that the average spectral peak frequency reached about 0.69 cpd during the eddy period. This value is very close to our observed  $\omega$  p.

**Suggested revision:**

The observed near-inertial spectral peak,  $\omega_p=0.667$  cpd, in the CE is consistent with the effective inertial frequency,  $f_eff=f+\zeta\approx 0.69$  cpd, estimated from surface geostrophic currents and the local inertial frequency.

Response: Thanks for this comment. We have rephrased as your advised: "The observed near-inertial spectral peak,  $\omega_{\rm p}$ =0.667 cpd, in the CE is consistent with the

effective inertial frequency,  $f_{eff} = f + \zeta \approx 0.69$  cpd, estimated from surface geostrophic currents and the local inertial frequency."

**3. The authors appear to have misunderstood my earlier concern.**

Near-inertial wind work (NIWW) refers to the rate of energy transfer from wind stress to near-inertial motions, and it should be computed as:

where both the wind stress  $\tau$  i and surface current u\_i are band-pass filtered to isolate near-inertial frequencies.

However, in Equation (6) (and accompanying text in lines 120-126), it seems that the wind stress is not filtered to retain only the near-inertial components. This means the resulting calculation does not represent near-inertial wind work.

I recommend that the authors carefully check their calculation, ensure appropriate filtering is applied to the wind stress, and revise the corresponding interpretation accordingly.

Response: Thanks for this comment. We reviewed several different references and found that some of them directly used wind stress for calculating the near-inertial wind work (Liu et al., 2019; Voelker et al., 2020), while others performed calculations after applying near-inertial filtering to wind stress (Flexas et al., 2019; Voet et al., 2024a). Relevant references are listed at the end. After careful consideration, we concluded that using filtered wind stress is more appropriate and thus have made corresponding adjustments. The change is as follows:

**Figure 5: (a)** Time series of raw (thin lines) and daily-smoothed (thick lines) averaged wind-input NIKE at Q1-Q4. **(b)** Time series of raw (thin lines) and daily-smoothed (thick lines) area integrated EKE.

**section 2.2:**

Near-inertial wind work was estimated as (Alford, 2001; Dasaro, 1985; Voet et al.,

2024b):

$$W = \vec{\tau}_i \cdot \vec{u}_i, \tag{6}$$

where  $\vec{u}_i$  is near-inertial velocity at sea surface derived from CMEMS products and  $\vec{\tau}_i$  is bandpassed near-inertial wind stress after calculated as (Alford, 2020; Liu et al., 2019):

**section 3.2:**

The wind work during the eddy period is about several orders of magnitude smaller than that affected by typhoons (Ouyang et al., 2022; Yuan et al., 2024) and several times smaller than the wind work results of Voet et al. (2024) in the Iceland basin.

**4. Typos:**

L226: qualified -> quantified

L237: Q1 -> Q1, Q2?

**Response:** Thanks for this comment. We have changed these two typo mistakes.

**References:**

Flexas, M. M., Thompson, A. F., Torres, H. S., Klein, P., Farrar, J. T., Zhang, H., and Menemenlis, D.: Global Estimates of the Energy Transfer From the Wind to the Ocean, With Emphasis on Near-Inertial Oscillations, Journal of Geophysical Research: Oceans, 124, 5723-5746, 2019.

Liu, Y., Jing, Z., and Wu, L.: Wind Power on Oceanic Near-Inertial Oscillations in the Global Ocean Estimated From Surface Drifters, Geophysical Research Letters, 46, 2647-2653, 2019.

Voet, G., Waterhouse, A., Savage, A., Kunze, E., MacKinnon, J., Alford, M., Colosi, J., Simmons, H., Klenz, T., Kelly, S., Moum, J., Whalen, C., Lien, R.-C., and Girton, J.: Near-Inertial Energy Variability in a Strong Mesoscale Eddy Field in the Iceland Basin, Oceanography, doi: 10.5670/oceanog.2024.302, 2024a. 2024a.

Voet, G., Waterhouse, A. F., Savage, A., Kunze, E., Mackinnon, J. A., Alford, M. H., Colosi, J. A., Simmons, H. L., Klenz, T., and Kelly, S. M.: NEAR-INERTIAL ENERGY VARIABILITY IN A STRONG MESOSCALE EDDY FIELD IN THE ICELAND BASIN, Oceanography, 37, 2024b.

---

## Author Response (AR4)

MS No.: egusphere-2025-283

Title: Enhancement of near-inertial waves by cyclonic eddy in the northwestern South China Sea during spring 2022

**Point-by-Point Response to Editor**

**Specific Comments**

The paper can be published when the point below, is also acknowledged in the manuscript. This is needed for other to be able to reproduce results or provide appropriate comparison.

The point below is from the response to the reviewer.

"The moving average needs to be applied after multiplying the variables; however we applied the moving average to individual variables before the multiplication, which thus resulted in excessive smoothing of the final results."

**Dear Editor.**

Thanks for the comment. The above response just reexplained the mistake we made in the first version in which we used wrong time-window to do time-smooth. The right figures and results regarding this point were already corrected in the second version, thus other researchers can reproduce the results and provide appropriate comparisons. Thanks again.